# Nomenclatural Type Identification of Names in North African *Tamarix* (*Tamaricaceae*)

**DOI:** 10.3390/plants12233969

**Published:** 2023-11-25

**Authors:** José Luis Villar, María Ángeles Alonso, Manuel B. Crespo, Mario Martínez-Azorín

**Affiliations:** Department of Environmental Sciences and Natural Resources (dCARN), University of Alicante, P.O. Box 99, 03080 Alicante, Spain; ma.alonso@ua.es (M.Á.A.); crespo@ua.es (M.B.C.); mmartinez@ua.es (M.M.-A.)

**Keywords:** *Tamarix*, *Tamaricaceae*, nomenclature, typification, Mediterranean basin, halophytes

## Abstract

*Tamarix* is native to Eurasia plus the northern and southern territories of Africa, with some species being introduced into America and Oceania. They are usually found in arid, desertic, or subdesertic areas, often on saline or subsaline soils, in Mediterranean, temperate, or subtropical climates. The genus is renowned for its complex taxonomy, which is usually based on rather variable or unstable characters, which leads to contrasting taxonomic treatments. As part of the taxonomic revision of *Tamarix* undertaken by the authors, ten names (i.e., *T. africana, T. bounopoea*, *T. brachystylis* var. *fluminensis, T. malenconiana*, *T. muluyana*, *T. tenuifolia*, *T. tingitana*, *T. trabutii*, *T. valdesquamigera,* and *T. weyleri*) published from material collected in the southwestern parts of the Mediterranean basin are taxonomically and nomenclaturally discussed after analysing their original material. Eight intended holotypes are corrected here to lectotypes; one epitype is designated for *T. africana* to warrant current use of the name; and one isotype, 30 isolectotypes, and 11 syntypes are also identified for the studied names. Further, the taxonomic identity of all names and their eventual synonymic placement are accordingly discussed.

## 1. Introduction

*Tamarix* L. (Tamaricaceae Link) is a genus native to Eurasia, occurring from the Atlantic European coasts to the Pacific Asian ones, which is also present in both the northern areas and the southernmost territories of Africa. Due to human activities, some species spread into other parts of the globe, and became invasive in America and Oceania [1,2]. Although some authors estimate between 60 and 90 species in the genus [3,4,5,6,7], more recently, Villar et al. [2] considered between 70 and 75 species as more reliable numbers. Taxa of *Tamarix* are usually halophytic tall shrubs inhabiting arid, desertic, or subdesertic areas under Mediterranean, temperate, or subtropical climates [7]; there, they can constitute the potential shrubby vegetation of saline soils [8].

Most of the authors who dealt with *Tamarix* considered the taxonomy of the genus to be complex [3,4,7,9,10], since many species were described based on morphological characters that later revealed unstable or highly variable, even in a single individual [11,12,13]. In consequence, scarce consensus exists on the delimitation of *Tamarix* species in different taxonomic treatments over time. Further, hybridisation can occur between taxa that are morphologically quite distinct [14,15] and those processes must have played a key role in the evolution of the genus [2]. The taxonomic and nomenclatural contributions by Baum [3,4] much improved the knowledge of *Tamarix* in its distribution area, stressing the importance of some characters, such as the staminal disc morphology or papillosity of the inflorescence rachis, among others. However, recent integrative research combining morphological, molecular, and biogeographical data has shown that even those characters, often accepted as diagnostic, exhibit indeed a wider plasticity [2]. These points to a broader circumscription of species and different synonymic relationships among them, not fitting entirely with Baum’s traditional arrangement [16].

In this scenario, the study of the type material is necessary to fix the use of names described in *Tamarix*, so that botanists can produce consistent decisions on taxonomy of the genus [17]. Some authors have made effective typification of *Tamarix* names in the past [18], especially Baum [3,4] who compiled almost every name published to that time and referred to the types of each one. However, as stressed in recent works [17,19,20], some of these type indications required some technical corrections according to the nomenclatural rules of the *Shenzhen Code*, hereafter abbreviated as ICN [21].

In the southwestern part of the Mediterranean basin (i.e., the Iberian Peninsula and the Maghreb), *Tamarix* is currently accepted to be represented by 10 species [16,22]: *T. africana* Poir., *T. amplexicaulis* Ehrenb., *T. aphylla* (L.) H.Karst., *T. boveana* Bunge, *T. gallica* L. (*T. canariensis* auct.), *T. macrocarpa* (Ehrenb.) Bunge, and *T. passerinoides* Desv. (native species), plus *T. chinensis* Lour., *T. parviflora* DC., and *T. ramosissima* Ledeb. (aliens). None is exclusive to that territory, and they are all spread into other parts of the Mediterranean basin. Several additional taxa were described from that area in the first half of the 20th century, mainly by Jules Battandier, René Maire, Carlos Pau, Frère Sennen, and their collaborators. Most of them were previously synonymised with any of the above-mentioned species accepted here, and their types were partly discussed elsewhere [3,4,16,17,18,19,20]. Recent references to taxa native to the eastern Mediterranean basin and southwestern Asia, such as *T. arborea* (Ehrenb.) Bunge, *T. arborea* var. *subvelutina* Boiss., *T. mannifera* Ehrenb. ex Bunge, *T. meyeri* Boiss., *T. octandra* Bunge, *T. rosea* Bunge, or *T. tetragyna* Ehrenb. [23,24,25,26], have been properly discussed and finally disregarded in the studied area [16].

Continuing with our taxonomic studies on *Tamarix*, in the present contribution we discuss the taxonomy and nomenclature of ten names published for plants occurring in the southwestern territories of the Mediterranean basin, mainly in North Africa. Eight intended holotypes are corrected here to lectotypes, one epitype is designated for *T. africana* to warrant the current use of that name, and one isotype, 30 isolectotypes, and 11 syntypes are identified for some of the concerned names. The taxonomic identity is commented on for all names, and their synonymic placement is confirmed or discussed accordingly.

## 2. Materials and Methods

Herbarium material and digital images of specimens from ABH, BC, FI, G, H, K, LY MPU, P, VAL, and W (acronyms according to Thiers [27]) were studied. Other data were also retrieved from diverse internet sources, such as GBIF [28], JACQ [29], JSTOR [30], NHM [31], PARLATORE [32], or ReColNat [33]. Authorities of plant names follow IPNI [34], though other nomenclatural databases, such as POWO [35] and APD [36], were checked for verification. Barcode numbers are placed after the corresponding herbarium acronym when available. The names analysed are numbered consecutively and listed alphabetically. For nomenclatural purposes, labels of types are transcribed literally, as in the herbarium sheets, and between inverted commas, leaving our own notes or comments aside. Typewritten words on labels have maintained the standard form, whereas handwritten text has been converted to italics. Names previously treated as synonymous with *T. canariensis* Willd. are now included in *T. gallica* following Villar [22], since the circumscription of the former name is regarded here in a narrower sense that applies only to plants from the Canary Islands [2,16]. Nomenclatural issues conform to the articles of the ICN [21].

## 3. Results and Discussion

### 3.1. Tamarix africana Poir., Voy. Barbarie 3: 139. 1789

*Type*: [ALGERIA]. “*Cotê de barbarie. Poiret*” (indicated as “holotype” by Baum [3] and corrected to lectotype by Villar et al. [11]: P-LA00287249! from Herbier de Lamarck; isolectotype: P00166702! [Label 1: “*Tamarix africana. Floribus pentandris confertissimis, spicis crassis, brevibus.”*; Label 2: “*Tamarix africana (n.) ex Numidia”*; Label 3: Herb. Poiret in Herb. Moquin-Tandon]; **epitype designated here:** P05113423! [Algérie. Dèp. Alger: Bas-fonds humides, au pied des collines du Sahel, au Sud de Koléa, 23 mars 1960, *A. Dubuis & L. Faurel 3815*].

*Tamarix africana* (Figure 1a,b) is one of the most common species within the genus along its western Mediterranean distribution [11]. The taxonomic identity and phylogenetic relations of the species are clear overall, with some discussion about the recognition of *T. africana* var. *fluminensis* (Maire) B.R.Baum (*T. brachystylis* var. *fluminensis* Maire) as a rightful taxon [2,4,11,37] (see comments below for that variety). The name *T. africana* was proposed by Poiret [38], and the description was based on materials collected in his trip through the northwestern African coast (ancient Numidia or Barbary, partly including the current territory of Morocco, Algeria, Tunisia, and western Libya) in 1785 and 1786. Although no precise location was cited in the protologue, Poiret [39] later noted in Lamarck’s *Encyclopédie Méthodique*: “J’ai découvert cet arbrisseau dans la Barbarie, aux environs d’Hippone, proche les bords de la mer. M. Desfontaines l’a également recueillie aux environs d’Alger.” [I have found this shrub in Barbary, in the surroundings of Hippone—currently Annaba, Algeria—, near the seashore. Mr. Desfontaines has also gathered it around Algiers]. Therefore, any North African material collected in that area and belonging to Poiret’s herbarium or being collected by him, might be considered as original material.

Baum [3,4] stated that the “holotype” was a specimen by Poiret kept at Lamarck’s Herbarium in the MNHN, Paris (P00287249). Moreover, Baum also mentioned two isotypes from Cosson’s herbarium and Moquin-Tandon’s herbarium, both preserved at P. The authors visited the Paris herbarium several times since 2013 and were unable to locate Cosson’s voucher. The specimen kept at Moquin-Tandon collection (P00166702) is part of the former Poiret’s herbarium and bears a fragment morphologically and phenologically resembling those in the lectotype specimen kept at Lamarck’s herbarium, and probably they all are part of a sole gathering. Therefore, Baum’s mention of holotype is to be corrected to effective lectotype designation (Art. 9.10 of the ICN) as suggested by Villar et al. [11], and the specimen at Moquin-Tandon herbarium can be considered an isolectotype (a true duplicate of the lectotype). Moreover, due to their age, the state of conservation of both existing specimens is not good enough to serve as proper references for the name since they lack the minimum features necessary for identification and to warrant a precise application of the name. The specimen from Lamarck’s herbarium is represented by a single branch that bears only a few leafed twigs and a couple of broken racemes with no complete flowers on them. The specimen at Moquin-Tandon’s is in a similar state, but it includes a small envelope with a few racemes. Moreover, the syntype is affixed to the right side of the herbarium sheet, with the left corresponding to a different specimen. Under these circumstances, we consider that the designation here of an epitype (Art. 9.9 of the ICN) on the specimen P05113423 (Figure 1a), an Algerian collection from near Algiers that matches the current concept of the species, will provide stability for further use of the name *Tamarix africana.*

### 3.2. Tamarix bounopoea J.Gay ex Batt. in Batt. & Trab., Fl. Algérie, Dicot.: 321. 1889

*Type*: [ALGERIA]. “B. BALANSA, PL. D’ALGERIE, 1852. 671. TAMARIX DESERTI (Boiss. Diagn. Pl. or. X, p. 9) Var. (J. Gay) Bords du *Chott-el-Chergui*, près de *Khrider*, cercle de *Saïda.* 30 mai.” (indicated as “holotype” by Baum [3], and **corrected here to lectotype**) K000242686 [digital image!]; isolectotypes: G00015657!, G00015658!, MPU008075!, MPU008077!, W1889-66955!). Syntypes: “B. BALANSA, PL. D’ALGÉRIE, 1853. 989. TAMARIX BOUNOPOEA, Gay in Coss. Rapp. Alg. 1852. T. deserti var. Gay in Balansa alg. exicc., 1852, nº 671. (J. Gay.) Environs de *Biskra.* Commencement d’avril.” (MPU008076!, MPU008078!, LY0130366 [digital image!], P05171516!).

Battandier [40] rescued and validated this unpublished name by John Gay, which was taken into consideration by later authors such as Munby [41,42]. We have not been able to check Gay’s original manuscript, which is kept at Kew Main Library. Battandier and Trabut’s description mentions three sites with not much detail on dates or collection numbers, but mentioning three localities: Le Kreider, Biskra, and Tunisia. Baum [3], who did have access to Gay’s manuscript, stated that the holotype was a specimen at K, belonging to the *Balansa 671* gathering, and mentioned other copies present at E, G, O, and W. The only specimen found at K (K000242686; Figure 2a) is corrected here to lectotype (Art. 9.10 of the ICN), and all other specimens belonging to *Balansa 671* are therefore isolectotypes. Provided that the valid publication was accomplished by Battandier [40], materials collected around Biskra in 1853 and belonging to the gathering *Balansa 989* can be considered as syntypes. There is an issue with the right spelling of *T. bounopoea.* In some taxonomical databases [34,35,36], it appears as “*Tamarix bounopaea*”. Others [28,33] admit double entries, either with “a” or “o”, which make it difficult to locate the specimens. Both the original publication by Battandier [40] and the labels on the original materials use the spelling “*Tamarix bounopoea*”. Therefore, we consider this latter variant to be the original and correct spelling to be used onwards. The name *T. bounopoea* is currently accepted as a synonym of *T. boveana* Bunge [43].

### 3.3. Tamarix brachystylis var. fluminensis Maire in in Bull. Soc. Hist. Nat. Afrique N. 26: 192. 1935

Type: [MOROCCO]. “*Dr. R. Maire–Iter maroccanum XXIV. Socio D^re^ E. Wilczek 1934. Tamarix brachystylis J. Gay var. fluminensis Maire. Secus amnes Anti-Atlantis in valle Içafen inter Igherm et Akka, 1100 m, 2536 cf. praeparata, 9-4-1934*” (indicated as “holotype” by Baum [3], and **corrected here to lectotype**): P00166718!; isolectotypes: MPU003366!, though the date is shown “die 9 aprilis”; RAB013506 [digital image!], though data on the label are presented in a rather different way: “Dr. R. MAIRE–Iter maroccanum XXIV. Socio D^re^ E. WILCZEK 1934. *Tamarix brachystylis J. Gay var. fluminensis Maire. Secus amnes vallis Içafen infra Igherm Anti-Atlantis, 1100 m,* die *9* aprilis”.

This name was published by Maire [44], including the sentence “Hab. secus torrentes Anti-Atlantis: prope Içafen (Flumina) inter Igherm et Akka, ad alt. c. 1100 m, aprili florens”. [it grows along ravines in the Anti-Atlas: near Içafen (rivers) between Igherm and Akka, at ca. 1100 m elevation; it flowers in April]. In the protologue, the new variety was compared to both *T. battandieri* Trabut and *T. boveana* Bunge on account of its muticous anthers. However, the very long bracts overtopping calyx and the 10-lobed disc clearly differentiated both taxa from *T. brachystylis* var. *fluminensis*. Baum [3,4] treated Maire’s taxon as *T. africana* var. *fluminensis* (Maire) Baum, and separated it from the typical *T. africana* by the “racemes narrower and longer than in var. *africana*, 6–8 cm long, 5 mm broad, densely flowered; petals obovate; bracts usually exceeding calyces.” The “holotype” of the name was said to be housed at P, with an “isotype” at RAB. However, we have traced three vouchers fitting with the indications in Maire’s protologue, which are conserved in Montpellier (MPU003366), Paris (P00166718), and Rabat (RAB013506) herbaria. All those specimens are duplicates of a collection in the exsiccata “Iter maroccanum XXIV. Socio D^re^ E. Wilczek 1934”. In this context, Baum’s mention of holotype is here corrected to lectotype (Figure 2b), and hence considered a valid lectotypification (Art. 9.10 of the ICN); further, the duplicates are to be treated as isolectotypes, despite minor differences in the label wording. Regarding the taxonomical relationships of *T. brachystylis* var. *fluminensis*, the type material is almost indistinguishable from the typical *T. africana*. The MPU specimen, which we could examine in detail, shows the following characters: (i) young branchlets and the rachis of racemes are papillate; (ii) racemes fall into the narrower width range of *T. africana* (approximately 5 mm), but they are not very long (shorter than 3.5 cm) as indicated in Baum’s description; (iii) flowers are pentamerous, with bracts approximately 2 mm long (not especially long for *T. africana*), triangular, not overtopping calyxes; (iv) sepals 1.2–1.5 × 0.7–1 mm (within the size range of the typical var. *africana*); and (v) petals 1.7–2.2 × 1–1.2 mm, elliptical in outline. However, because the racemes are still starting to bloom and most flowers are not entirely developed, which may affect measurements, some of the floral features argued for the separation of var. *fluminensis* might perhaps be disregarded as diagnostic. Therefore, until further studies show stronger evidence, we consider this variety to fall into the synonymy of *T. africana.*

### 3.4. Tamarix malenconiana Maire in Bull. Soc. Hist. Nat. Afrique N. 35: 194. 1935

Type: [MOROCCO]. “DR. MAIRE–Iter maroccanum XXIV. Socio D^re^ E. WILCZEK 1934. Tamarix Malenconiana Maire. Ad ripas fluminis Drâa prope Zagora. Leg. G. Malençon. die 23 februarii” (indicated as “holotype” by Baum [3], and **corrected here to lectotype**): P00166729!; isolectotype: MPU003361!).

*Tamarix malenconiana* was described by Maire [44] on materials collected by G. Malençon near Zagora (Morocco). Baum [3] stated that the “holotype” of the name was kept at P and included that specimen under the studied material of *T. africana.* There is indeed a specimen (P00166729; Figure 3a) that matches Maire’s protologue. However, the presence of a duplicate specimen at MPU (MPU003361) means that Baum’s indication is to be considered a lectotype designation, and it is accordingly corrected here (Art. 9.10 of the ICN). After examining the available original material, we have some doubts about the taxonomic placement of *T. malenconiana.* The floral parts fall into an intermediate range between *T. africana* and *T. gallica.* Moreover, the collection date and phenology do not match with the aforementioned species. There are still several features we have not been able to observe in the field on specimens from so far south of Morocco. There is scarce information regarding whether those specimens are evergreen or deciduous, and the extant phylogenetic information [2] did not include specimens collected around that area. Therefore, we prefer not to provide a taxonomical adscription to this name until new observations on living materials or more complete collections are available.

### 3.5. Tamarix muluyana Sennen, Diagn. Nouv.: 189. 1936

*Type*: [MOROCCO]. “1933.- PLANTES D’ESPAGNE.- F. SENNEN. Nº 8784. Tamarix muluyana Sennen. Maroc: Ulad-Settut, bords du Muluya. 12-X. Leg. Hno. Mauricio” (indicated as “holotype” by Baum [3], and **corrected here to lectotype**): MA79042!; isolectotypes: BC137946 [digital image!], BM000629770!, G00015652!, H1380512 [digital image!], MPU009320!, P00166732!, RAB008049 [digital image!], VAL157910!, W1934-7864!).

The name *Tamarix muluyana* was first published without a description or diagnosis [45], and therefore it was a *nomen nudum* (Art. 38 of the ICN). However, three years later, Sennen [46] validated the name by providing a proper diagnosis. Baum [3] stated that the “holotype” was housed in Madrid (MA), and cited isotypes at BM, G, P, and W. Nevertheless, the protologue [46] did not refer to a single voucher, but to an entire gathering distributed in the exsiccata “Plantes d’Espagne–F. Sennen–1933”, numbered 8784: “Hab.–Maroc: Ulad-Settut, rives du Muluya. Leg. Hno. Mauricio”. Provided that vouchers of the same collection are scattered in different herbaria, the only specimen found in MA (MA79042; Figure 3b) should be considered not a holotype but a lectotype designated by Baum [3], which is accordingly corrected here (Art. 9.10 of the ICN). We consider *T. muluyana* as a perfect example of an autumnal bloom of *T. gallica*, which is congruent with Sennen’s comment in the protologue: “gr. *gallica*?”

### 3.6. Tamarix tenuifolia Maire & Trab. in Bull. Soc. Hist. Nat. Afrique N. 25: 296. 1934

*Type*: [ALGERIA]. “D^r^ R. MAIRE–Itinera Algerica 1933. *Tamarix tenuifolia Maire et Trabut. In Saharae septentr. ditione Oued Rhir, prope El_Arfiane in salsuginosis. 19-3*.” (indicated as “holotype” by Baum [3], and **corrected here to lectotype**): P00166709!; isolectotypes: BC137954 [digital image!], RAB013568 [ReColNat data (not seen)], P00166711! [“*D^r^ R. MAIRE–Itinera algerica.* HERBIER DE L’AFRIQUE DU NORD *Tamarix tenuifolia Maire et Trabut. In ditione Oued Rhir, prope el Arfiane in salsuginosis corolla alba 19-3-1933.*”]). Syntypes: “UNIVERSITÉ D’ALGER. HERBIER DE L’AFRIQUE DU NORD. *Tamarix tenuifolia Maire et Trabut (Typus). Sahara: Oued Rhir, dunes salées a El Arfiane. Leg D^r^ L. Trabut nº 2394. 20.3.1918. Dr R Maire.*” (MPU003236!, P00166710!); “(*Tamarix tenuifolia Trabut) El Arfiane 20 mars 1928* [sic]*. 2394*” (MPU003237!, MPU003238!).

In the protologue of *Tamarix tenuifolia*, Maire [47] wrote about the species: “Hab. in salsuginosis dittionis Oued Rhir Saharae algeriensis, prope El-Arfiane! (TRABUT)” [it grows in the saline marshes of Oued Rhir in the Algerian Sahara, near El-Arfiane! (Trabut)]. Maire also explained that he was able to study living specimens himself at that locality in 1933, which convinced him that *T. tenuifolia* was a rightful species. There are specimens at several herbaria from at least two gatherings that can be considered as original material: (i) the collection *Trabut 2394* made in March-1918 that matches Maire’s habitat indication, though it was not explicitly cited; and (ii) the gatherings made by Maire at the type locality in 1933 that surely correspond to those living specimens referred to in the protologue. Baum [3] stated that the “holotype” was one of the Maire’s specimens collected in 1933, of which he transcribed the label information, and isotypes were cited to be present at FI and RAB. Although two specimens kept at P belong to Maire’s 1933 gathering; P00166709 (Figure 4a) can be traced as the one mentioned by Baum [3] by literal transcription of its label, which shows slight differences regarding the one on P00166711. Provided that there are several syntypes and no clear indication of a precise specimen by Maire, Baum’s holotype mention is here corrected to lectotype designation (Art. 9.10 of the ICN). Other copies of that gathering must be considered isolectotypes, and specimens belonging to *Trabut 2394* are to be regarded as syntypes. It is worth mentioning that two of those syntypes at Montpellier (MPU003237, MPU003238) bear a smaller label showing a different collection year, March-1928, which is considered here a transcription mistake for March-1918. *Tamarix tenuifolia* can be found in some taxonomic treatments as a synonym of *T. passerinoides* [3,4,22]. However, it belongs to a group of taxa with amplexicaul leaves and 10-stamened flowers that are in need of deeper taxonomic revision to clarify the real number of different species involved.

### 3.7. Tamarix tingitana Pau in Mem. Soc. Esp. Hist. Nat. 12: 293. 1924

*Type*: [MOROCCO]. “VIAJE BOTÁNICO POR LA MAURITANIA por C. PAU.- Abril y Mayo 1921. Comisión de la Real Sociedad Española de História Natural. *Tamarix tingitana Pau. De Tanger a Fondak. 2 mayo*” (holotype: MA78992!; isotype: MPU008862! “UNIVERSITÉ D’ALGER HERBIER DE L’AFRIQUE DU NORD *Tamarix tingitana Pau fragmentum typi M. Tanger Leg C. Pau*”; Label 2: “*Tanger a Begazen 2 mayo 1921 C. Pau*”).

*Tamarix tingitana* was described by Pau [48] from material collected in northern Morocco. The protologue only includes a location reference indicating “De Tánger al Fondak” [from Tangier to El Fondak]. Only one specimen with that information on the label has been found (MA78992; Figure 4b). Provided that no other materials are available, it is acceptable that this is the specimen upon which Pau based his description, and, therefore, it should be considered as the holotype of *T. tingitana,* as Baum [3] already pointed out. However, there is a specimen in the Montpellier Herbarium (MPU008862) that is supposed to contain a fragment of the type. The collection date and the collector information match with those in the holotype; however, the locality differs slightly and does not mention El Fondak but “Begazen”. At that time, El Fondak was a spot on the route between Tangier and Tétouan, but we have not been able to locate Begazen. If we assume that the fragment at MPU008862 was taken from the holotype (“fragmentum typi”, as shown on the accompanying label), we have to consider that this voucher is an isotype. *Tamarix tingitana* is currently regarded as a synonym of *T. africana* [3,22].

### 3.8. Tamarix trabutii Maire in Bull. Soc. Hist. Nat. Afrique N. 22: 35. 1931

Type: [ALGERIA]. “D^r^ R Maire–Iter Saharicum. HERBIER DE L’AFRIQUE DU NORD. Tamarix trabutii Maire. In montibus Emmidir (Mouydir) Haci-el-Kheneg, in alveo amnis. 310 m. 28-2-1928. nº 253” (indicated as “holotype” by Baum [3], and **corrected here to lectotype**): P00166707!; isolectotypes: FI000642 [digital image!], MPU002363!, MPU002364!, RAB008000 [not seen]).

Maire [49] described *T. trabutii* explicitly, citing a precise gathering (*Maire 253*). Baum [3] stated that the “holotype” was kept at P, with isotypes at FI, RAB, and US. We have traced two additional vouchers belonging to *Maire 253* at MPU. Provided that no specific voucher among the original material was designated by Maire [49], Baum’s indication of “holotype” is corrected here to a valid lectotype designation (Art. 9.10 of the ICN) on the voucher P00166707 (Figure 5a). Regarding the taxonomic identity of *T. trabutii*, it is usually considered a synonym of *Tamarix amplexicaulis* [4,22]. However, we would keep our reservations since deeper taxonomical and phylogenetic studies will deal with the 10-stamened and amplexicaul-leaved group already commented on the above entry of *T. tenuifolia*.

### 3.9. Tamarix valdesquamigera Sennen, Diagn. Nouv.: 125. 1936

*Type*: [MOROCCO]. “1931–PLANTES D’ESPAGNE.- F.SENNEN. Nº 7846. Tamarix valdesquamigera Sennen. grex gallica. Maroc: Lit et marges du Nékor, Route d’Alhucemas. 7-VII. Leg. Sennen et Mauricio.” (indicated as “holotype” by Baum [3], and **corrected here to lectotype**): MA78835!; isolectotypes: BC137955 [digital image!], G00015654!, MPU008382!, MPU008383!, VAL157907!, W1933-5245!).

First published as a *nomen nudum* by Sennen and Mauricio [45], *Tamarix valdesquamigera* was validated by Sennen [46] with a description, the indication of a collection number (nº 7846), and a location described as “*Marges du Nekor et de l’Amekran, dans leur cours inférieur*”. On that basis, Baum [3] cited a specimen kept at MA (MA78835; Figure 5b) as the “holotype”, alongside isotypes kept at G and BM. All specimens mentioned by Baum, together with some others found by the authors during the last few years, are in fact syntypes. Therefore, Baum’s mention of holotype is corrected here to an effective lectotype designation (Art. 9.10 of the ICN). The remaining materials should be considered isolectotypes. *Tamarix valdesquamigera* is currently considered a synonym of *T. gallica* [22].

### 3.10. Tamarix weyleri Pau in Mem. Soc. Españ. Hist. Nat. 22: 293. 1924, “weylerii”

*Type*: [ALGERIA]. “VIAJE BOTÁNICO POR LA MAURITANIA por C. PAU.- abril y mayo 1921. Comisión de la Real Sociedad Española de Historia Natural. *Tamarix Weylerii Pau, p. 31. Tetuan; Río Martín, cerca del paso de la barca, margen derecha del río. Mayo*” (indicated as “holotype” by Baum [3], and **corrected here to lectotype**: MA78856!). Syntypes: “VIAJE BOTÁNICO POR LA MAURITANIA por C. PAU.- abril y mayo 1921. Comisión de la Real Sociedad Española de Historia Natural. *Tamarix Weylerii Pau, Pl de Yébala, 31. T. gallica Weyler, catálogo. Tetuan ad ripas fluminis, 8 Mayo*” (MA78854!); “VIAJE BOTÁNICO POR LA MAURITANIA por C. PAU.- abril y mayo 1921. Comisión de la Real Sociedad Española de Historia Natural. *Tamarix Weylerii Pau. Tetuan 15 Mayo*” (MA78855!); “UNIVERSITÈ D’ALGER. HERBIER DE L’AFRIQUE DU NORD. *Tamarix weyleri Pau. Fragmentum typi. M. Tétouan. Leg. C. Pau. Dr R. Maire*” (MPU008866!).

*Tamarix weyleri* was described [48], including the type locality: “Margen derecha del río Martín, antes de llegar a la barra [sic; most probably a typographic error for “barca”, which is indeed annotated on the label of the type material in Pau’s hand], en el camino de Beni Hozmar a Tetuán” [Right bank of Martín River, before the bar (sic; surely a mistake for ”boat”), on the way from Beni Hozmar to Tétouan]. Baum [3] regarded the specimen MA78856 (Figure 6) as the “holotype” of the name. That specimen was collected by Pau and shows a label with an almost perfect match with the location described in the protologue. However, there are some other specimens at MA that were collected by Pau around the same date (May-1921) in Tétouan and in the Martín River, and labelled by himself as *T. weyleri.* Moreover, a small fragment of the type material is preserved at MPU. All those specimens are to be considered syntypes, since it is almost sure that Pau used all of them for the description associated with *T. weyleri.* Therefore, Baum’s mention of holotype is corrected here to lectotype, and hence considered a valid lectotypification (Art. 9.10 of the ICN). *Tamarix weyleri* is currently considered a synonym of *T. gallica* [22].

## 4. Conclusions

Our research highlights the importance of careful examination of the protologues and type material of names in *Tamarix*, a genus in which many taxonomic and nomenclatural issues are still in need of revision. Considering the importance of types for the correct identification of taxa, which becomes even more important in taxonomically critical groups, in the present work, ten names of *Tamarix* taxa occurring mainly in the southwestern Mediterranean basin were investigated. In all cases, existing typification and synonymisation of the studied names were revisited, and some issues were found that needed clarification. Nomenclatural issues were fixed by correcting eight intended holotypes to lectotypes, as well as designating one epitype for *Tamarix africana*, which contributes to the nomenclatural stability of all concerned names. Previous synonymisation of some names was confirmed but was questioned or corrected in other cases. The results of the present paper aim, therefore, at contributing to the clarification of the systematic diversity of *Tamarix* and the stability of biological nomenclature, hence providing an additional base for further research on this complex plant group in the southwestern Mediterranean basin.

## Figures and Tables

**Figure 1 plants-12-03969-f001:**
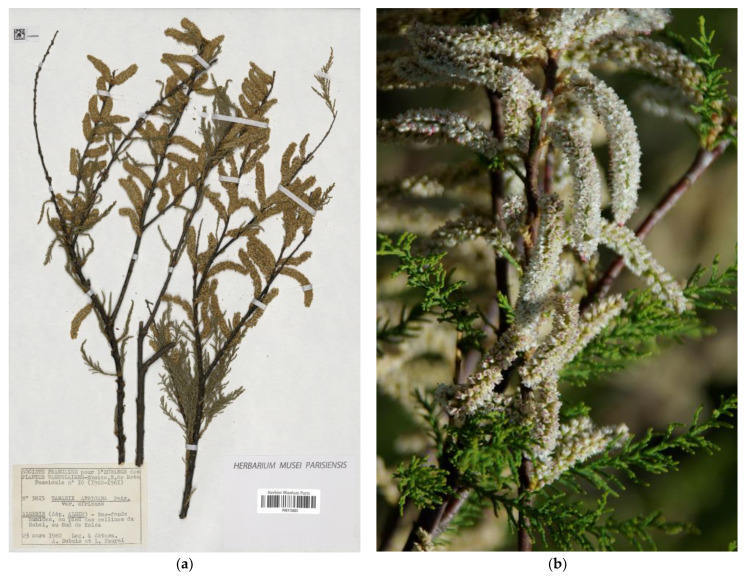
*Tamarix africana*. (**a**) Epitype designated here from near Koléa, Dept. of Algiers, Algeria (P05113243), reproduced with permission (© Muséum National d’Histoire Naturelle, Herbarium, Paris); (**b**) detail of a flowering plant from Aknoul, Morocco (photo: José Luis Villar, 21 April 2009).

**Figure 2 plants-12-03969-f002:**
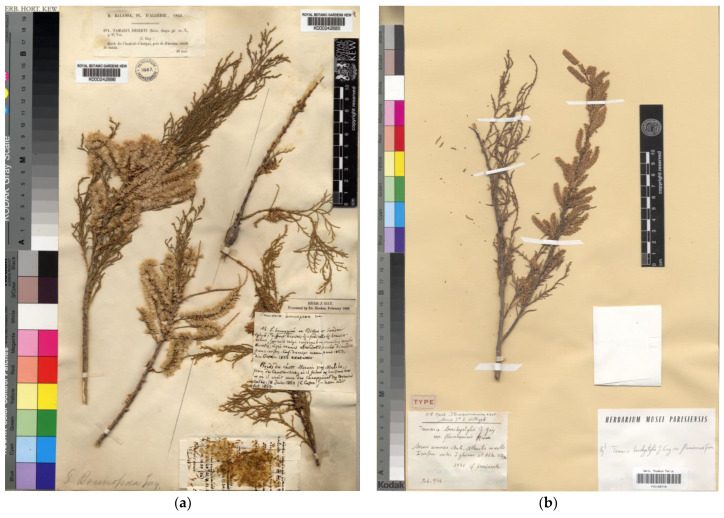
Lectotype of: (**a**) *Tamarix bounopoea* (K000242686), reproduced with permission (© The Royal Botanic Gardens, Kew); (**b**) *Tamarix brachystylis* var. *fluminensis* (P00166718), reproduced with permission (© Muséum National d’Histoire Naturelle, Herbarium, Paris).

**Figure 3 plants-12-03969-f003:**
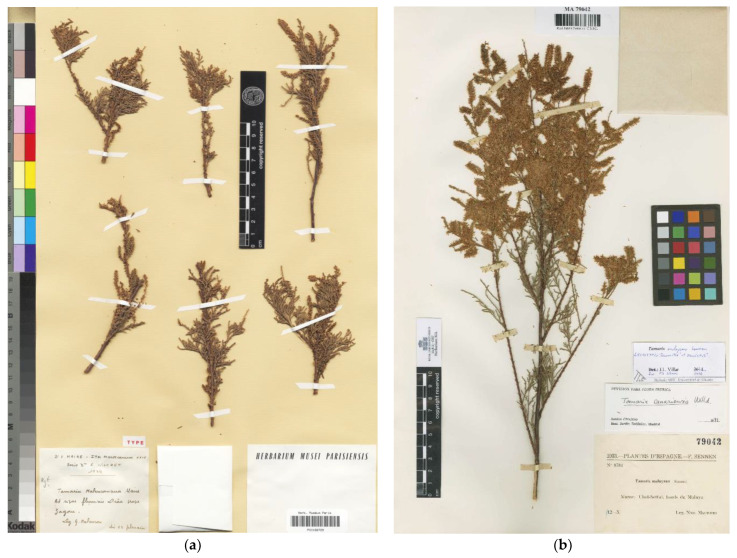
Lectotype of: (**a**) *Tamarix malenconiana* (P00166729), reproduced with permission (© Muséum National d’Histoire Naturelle, Herbarium, Paris); (**b**) *Tamarix muluyana* (MA79042), reproduced with permission (© Real Jardín Botánico, CSIC, Madrid).

**Figure 4 plants-12-03969-f004:**
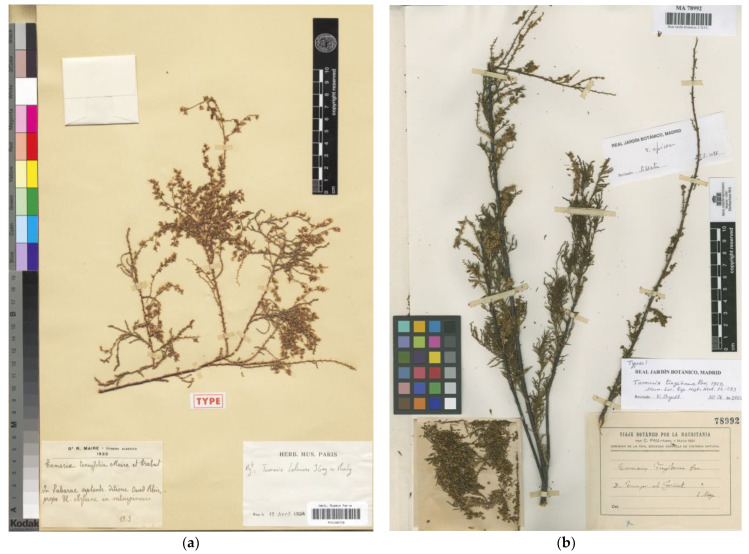
(**a**) Lectotype of *Tamarix tenuifolia* (P00166709), reproduced with permission (© Muséum National d’Histoire Naturelle, Herbarium, Paris); (**b**) holotype of *Tamarix tingitana* (MA78992), reproduced with permission (© Real Jardín Botánico, CSIC, Madrid).

**Figure 5 plants-12-03969-f005:**
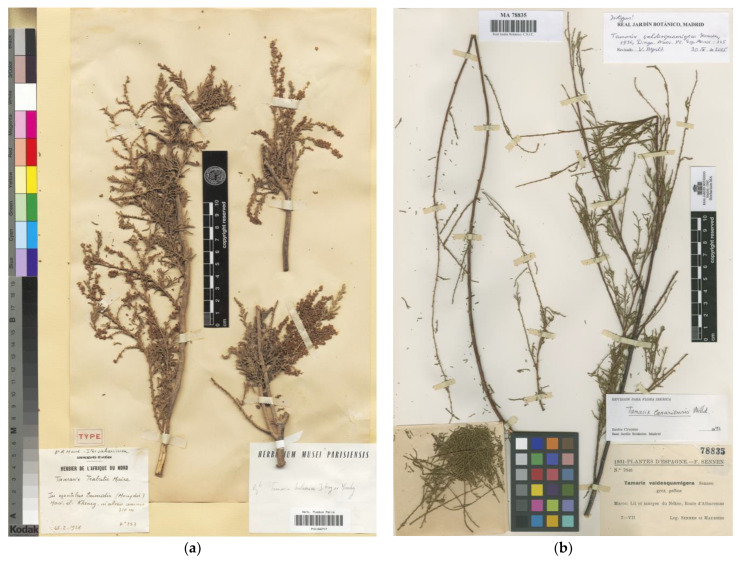
Lectotype of: (**a**) *Tamarix trabutii* (P00166707), reproduced with permission (© Muséum National d’Histoire Naturelle, Herbarium, Paris); (**b**) *Tamarix valdesquamigera* (MA78835), reproduced with permission.

**Figure 6 plants-12-03969-f006:**
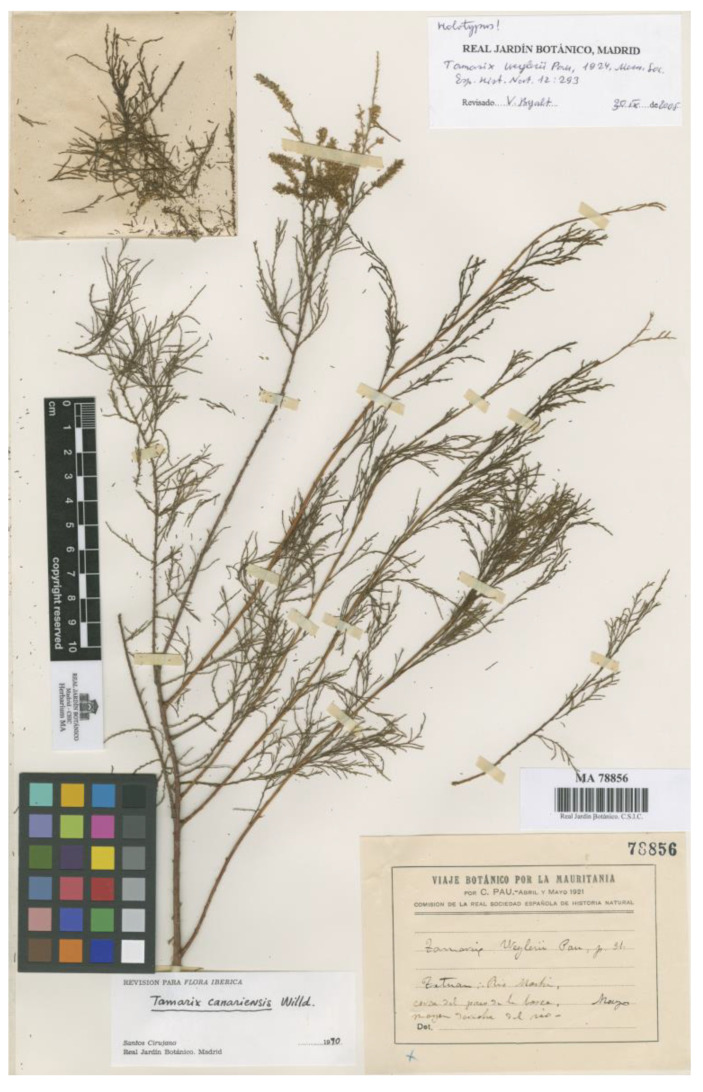
Lectotype of *Tamarix weyleri* (MA78856), reproduced with permission (© Real Jardín Botánico, CSIC, Madrid, Spain).

## Data Availability

Images of plant material cited in the text, when not included in figures, are mostly available at JSTOR Global Plants (https://plants.jstor.org/plants/browse, accessed on 12 October 2023) and the web pages of the concerned herbaria.

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
