# Peer review of "Nomenclatural Type Identification of Names in North African Tamarix (Tamaricaceae)"

_plants, 2023, doi:10.3390/plants12233969_

Round 1

Reviewer 1 Report

Comments and Suggestions for Authors

This manuscript constitutes a solid contribution to clarify nomenclatural and taxonomical issues on the complex genus Tamarix. The paper is consistent, well written (and includes usefuls and high quality figures) and it is highly valuable for botanists working on Tamarix and the mediterranean flora.

I do not raise any major concerns with the mnanuscript. However, I indicate a minor comments below:

 Is the name “T. brachystylis var. fluminensis Maire” [T africana var. fluminensis (Maire) B.R.Baum] typified? This variety has uncertain taxonomic value (see lines 97-99). It would be convenient, if possible, to provide some more taxonomic information about this variety.

Author Response

Dear Reviewer 1,

Many thanks for your report. Following your suggestion, we have added the case of Tamarix brachystylis var. fluminensis (pp. 5-6, plus a new figure "2b" of the lectotype), with details on typification and discussion on its taxonomic affiliation. This also makes clearer the understanding of the Tamarix africana case. I hope this addition is fine for you.

Best wishes, and thanks again.

The authors

Reviewer 2 Report

Comments and Suggestions for Authors

The manuscript deals with nomenclatural issues concerning some names in genus Tamarix from north Africa.

The topic is interesting and falls within the scope of the journal.

The presentation is very clear and well structured. English used is ok.

I strongly suggest adding a Conclusion paragraph (for this and other suggestions/corrections see the file in attachment).

After the suggested corrections (file in attachment), in my opinion the manuscript will deserve publication in Plants journal.

Comments on the Quality of English Language

The English used is quite good.

Author Response

Dear Rev2,

Many thanks for your kind message. We have read carefully your suggestions and have included most of them in the new version of the Ms, in which all changes/additions are now marked in yellow. There are, however, some aspects needing further comments, as follow:

(i) p. 1: We retained all keywords as in the original submission, since the journal seems to allow keywords already found in the title. The editor will be informed about this point for any additional change.

(ii) p. 3: "brachystylis" is correct and it was the spelling used in the protologue of the species. Further, the name "Tamarix brachystylis" is accepted in IPNI and other nomenclatural databases, so that we prefer to retain it.

(iii) p. 5: In the comment for Tamarix malenconiana, the sentence "There are still several features we have not been able to observe in the field on specimens that south of Morocco." has been fixed to: "There are still several features we have not been able to observe in the field on specimens from so far south of Morocco."

(iv) p. 10: We added a new section "4. Conclusions", as you suggested. We also used your sentences as a base from which we built our new text. However, we did not include any reference since they seemed no strictly necessary. Anyway, many thanks for the interesting contributions you made.

Further, we have made some changes in the text and finally added a new paragraph and six new references in the introduction (p. 1-2). More importantly, a new taxon (Tamarix brachystylis var. fluminensis Maire in in Bull. Soc. Hist. Nat. Afrique N. 26: 192. 1935) has been added following suggestions by other reviewers. We hope this will make the text clearer and will solve some uncertainioties concernin citation of that name in the section of Tamarix africana.

We appreciate very much your input, and are very happy with the revised taxt that is much improved.

Yours,

The authors